# FMNISCF: Fine-Grained Multi-Domain Network Interconnection Security Control Framework

**Bo Lu** [1,2]🄾**, Ruohan Cao** [1,3]**, Luyao Tian** [1,3]**, Hao Wang** [1,2] **and Yueming Lu** [1,3,*]

1   Key Laboratory of Trustworthy Distributed Computing and Service, Ministry of Education,
    Beijing University of Posts and Telecommunications (BUPT), Beijing 100876,
    China; bolu@bupt.edu.cn (B.L.); caoruohan@bupt.edu.cn (R.C.); tianluyao@bupt.edu.cn (L.T.);
    wanghao2018@bupt.edu.cn (H.W.)
2   The School of Cyberspace Security, Beijing University of Posts and Telecommunications (BUPT),
    Beijing 100876, China
3   The School of Information and Communication Engineering, Beijing University of Posts and
    Telecommunications (BUPT), Beijing 100876, China
*   Correspondence: ymlu@bupt.edu.cn

**Abstract:** The integrated air-ground multi-domain network provides users with a set of shared infrastructures. Security policies can be defined flexibly in the context of multi-domain network semantics. The packet filter module in the security gateway can run efficiently, which is an urgent requirement in this network environment. The framework combined with multi-domain network semantics implements the transformation into rules. It replaces the traditional manual method of configuring rules. The framework supports the whole life cycle management of rules from generation state and distribution state to execution state. In the aspect of security, the map security and semantic security are analyzed and optimized, respectively. Finally, through a series of experiments, compared with iptables/DPDK-IPFW/BSD-IPFW/BSD-pfsense, the high efficiency of the scheme is verified.

**Keywords:** integrated air-ground multi-domain network; security interconnection gateway; security policy; security rule; semantic security

## 1. Introduction

In recent years, the complex network environment represented by the integrated air-ground network [1,2], IOT [3,4], and the complex private network [5,6] has developed rapidly, which has brought great convenience to people's daily life and also generated a lot of security related problems, especially those dealing with data security. Therefore, as an advanced technology to protect the legitimate access to data, the research on access control technology in complex network environment is particularly necessary.

The integrated air-ground network is multiple domain networks with a set of shared communication infrastructures. The differences among the securities of the multi-domain [7] require the communication in the application scenarios to have the complex adaptive ability. This adaptability includes the following aspects, that is, in the network, the business type/feature/security level changes with the spatiotemporal characteristics of multi-domain and multi-user. The integrated air-ground network can form physical or virtual network domains due to different user types, business types, and security levels. In order to ensure the security of the cross-domain communication, a security gateway needs to be set among the domains. We call this gateway the security interconnection gateway. The security interconnection gateway is a device that can connect the multi-domain networks

according to the security policies. It is set up in a domain interconnection node of isolated physical or virtual networks.

The complex network environment has the characteristics of dynamic device access, heterogeneous network segment, and frequent information flow across domains. For example, the integrated air-ground network is composed of heterogeneous networks such as space-based backbone networks, space-based access networks, ground-based node networks, ground Internet, and mobile communication networks. In this network, a large number of ground users have frequent dynamic access, and the real-time ground information obtained by the space-based access network flows to ground nodes via the space-based backbone network.

These characteristics bring many new requirements for access control technology in a complex network environment.

(1) fine-grained control: the complex network environment has a huge amount of information, and different users have different permissions to use the information, so that the coarse-grained control will bring lots of security problems.
(2) policy tracking: data information flows frequently between networks. If the corresponding control policy does not follow the data information ontology to the new network, users will lose control of the data.
(3) policy semantic normalization: data information flows across different networks, and inconsistencies in policy languages between networks may cause errors in policy transformation between networks.

Existing packet filtering techniques [8–12] were initially used in the network to provide security protection. The access control function of the network packet is implemented on the router, and the security function of it is separated. The working principle of the program of this technology is that in a interconnected network device, it works by loading some special instructions such as allowing or prohibiting packet transfer from specific source address, destination address, TCP terminal number, etc. Through the corresponding instructions for series of equipment inspections of packets, it can restrict dangerous packets in and out of the internal network. The advantages of the control are obvious. It can keep high transparency to users, and high work efficiency. However, he also has serious shortcomings. For example, the manager can't fully control information flow in the network. It only can control the most basic security maintenance. In the face of the high technical content of attack without resistance, it is difficult to upgrade maintenance, and so can't maintain the security of users.

These characteristics bring many new requirements for access control technology in a complex network environment. The fine-grained interconnection security control framework is a key technique for multi-domain interconnection and access control in multi-domain networks. The impacts of it are the following aspects:

First, for the complexity of rules working at the bottom, a policy needs to be designed to simplify the description of logical relationships in complex networks. With this abstract level of policy language, when designing business-oriented access control logic, network configuration scripts based on the policy language can look more concise and understandable. In engineering practice, the domain of the network can be easily configured, and the packet filter program can be executed efficiently in the gateway. The policy language designed for users in the network can be used to configure network security constraints flexibly and improve the readability and maintenance of programs. This requires normalization of policy semantics and data information flows across different networks. In addition, inconsistencies in policy languages between networks may cause errors in policy transformation between networks.

Second, in complex business application scenarios, inter-domain isolation needs to be supported. The traditional coarse-grained access control model can no longer fully meet the requirements of access control under complex networks in many scenarios. In recent years, fine-grained interconnection security control [13–15] has developed rapidly and has important applications in the internet of things security and cloud security.

Many of the current attacks are more and more application-oriented, such as cross-network scripting attacks, SQL injection attacks, and lunch attacks. This puts forward higher requirements on the dimension of data filtering. This requires the system to be able to filter packets at a fine-grained level. They include time and characteristics. The complex network environment has a huge amount of information, and different users have different permissions to use the information, so that the coarse-grained control will bring about lots of security problems. Data information flows frequently between networks. If the corresponding control policy does not follow the data information ontology to the new network, users will lose control of the data.

In addition, to turn access control policies for complex application scenarios into rules, a mapping needs to be designed. By designing a map, the polices are translated into the rules automatically. In the security policies, the description of multi-domain interconnection is supported. Users can flexibly set the network functions according to the network security requirements.

Finally, to meet the needs of high-speed network communication, it is necessary to design rules that can adapt to the high-speed filtering packet of security gateway and the fast filtering of gateway. According to our proposed rules, gateways can filter packets efficiently. By summarizing the above requirements, it can be concluded that the fine-grained interconnected security access control framework needs to design an abstract language (policy) that can directly describe complex application scenarios, with fine-grained attributes that resist more attacks. The gateway can load and execute rules efficiently. Mapping mechanisms can translate policies into rules. The research progress in these aspects will be introduced below.

### 1.1. Related Work

Due to the importance of fine-grained multi-domain interconnection security control to high-traffic network platforms and devices, various methods have been used to solve the problems related to multi-domain interconnection, conflict detection, rule transformation, and fine-grained control. In this section, we'll review existing technologies and describe the differences from FMNISCF. Regarding multi-domain interconnection management, Iizawa et al. [16] proposed a network orchestrator based on ODENOS (object-defined network allocation system) for orchestrating multi-layer and multi-domain networks. The network orchestrator can dynamically provide end-to-end paths according to requests of network services. Li et al. [17] proposed a controller cluster-based interconnecting framework based on load balance, which can be implemented through two levels of controller coordination, i.e., inter-cluster and intra-cluster interconnection. However, they have not carefully demonstrated and considered the security issues among domains. For the map from policies to rules, Min et al. [18] proposed a policy generation method for a control system in open environments. The main idea is to formalize the description of the control policy step by step, associate each stage of the policy generation in an automatic way, and reduce the difficulty of the description, generation, and final application of the policy. Hager et al. [19] proposed RuleBender, a rule set transformation technique that encodes decision tree search structures into the transformed rule set, which in turn can be traversed significantly faster. However, they have not demonstrated the security of rule map in detail, and there are more ways to utilize the traversal speed of rule set. Aiming at rule conflict detection, Pisharody et al. [20] proposed a security policy analysis framework Brew based on OpenDaylight SDN controller, which identified and solved cross-layer conflicts by using global priority ranking technology for convection rules in a decentralized environment. Khelf et al. [21] proposed an algorithm for dynamic detection of both intra and inter conflicts of an IPsec security policy. The proposed algorithm is based on a simple and comprehensive mechanism that uses Boolean functions to classify and identify. The resolution of intra-policy conflict is also integrated into the algorithm. However, there is no detailed discussion of the semantic security of the rules. For fine-grained management control, Liu et al. [22] proposed a fine-grained two-factor authentication (2FA) access control system for web-based cloud computing services, which requires a user key and a lightweight security device to complete fine-grained access control of the system and enhance the

security of the system. Pei et al. [23] proposed a fine-grained access control method for the cloud resources, and the basic idea is to use XACML as an access control language and optimize policies by data fragmentation and policy refinement algorithms. The XACML language can determine whether users can access web resources or not. Merindol et al. [24] proposed a new fine-grained multi-source measurement platform DCART. DCART is a distributed platform running over an operational network (RENATER, the French research and education network); it gathers data from several sources.

*1.2. Our Work*

However, the above systems do not introduce time features and cannot control the detailed content characteristics of the packet. Different from existing studies, our work generates rules by translating policies, and builds a tree structure of rules to filter packets. The same condition is integrated when the rules are generated by the policy, which ensures that there is no conflict and redundancy among rules. At the same time, the rules are arranged in lexicographical order, which facilitates binary search in packet filtering and improves matching speed. Our work introduces fine-grained control of time, characteristic, and adds constraint conditions of time and characteristics to the rules. When filtering packets, not only the five-tuples need to be matched, but also the time and the URL characteristics need to be matched. We also design the syntax of policies and rules respectively and discuss the semantic security of policy-to-rule transformations in multi-domain networks by using a formal method. In the experimental part, we compare the performance of our scheme with the netfilter/iptables of the Linux operating system. The results show that our scheme is more efficient.

## 2. Framework

In order to facilitate the management and application, a security policy language is designed, which fits users' usage habits. However, this security policy language has low performance and complex logic, which cannot be directly applied to high-throughput security interconnection gateway. Therefore, a map method is proposed, which interprets security policies as security rules.

Consider a situation where user $u_1$ of domain $d_1$ and user $u_2$ of domain $d_2$ can communicate with each other using $app_1$. The communication time is limited from 9:00 a.m. to 10:00 a.m. every day, and there is no restriction on feature contents. The details of $app_1$ communication among domains $d_1$ and $d_2$ are stored in a relationship and do not appear directly in the policy script. We can express it as follows:

$$\begin{bmatrix} default & : & drop \\ whitelist & : & \{d_1 - d_2, app_1\} \end{bmatrix}.$$

By means of map components, we can translate this policy into rules that can be loaded by the gateway. See the following:

$$\begin{bmatrix} 10.0.0.1\text{--}10.0.0.1, 20.0.0.2\text{--}20.0.0.2, 6\text{--}6, \\ 0\text{--}65{,}535, 23\text{--}23, 287{,}485{,}200\text{--}287{,}488{,}800, \_ : accept \end{bmatrix},$$

where 10.0.0.1 is the address of user $u_1$, 20.0.0.2 is the address of user $u_2$, 6 is the type of TCP, 0–65,535 is the source port range, 23–23 is the destination port of range, 287,485,200–287,488,800 means 9:00 a.m.–10:00 a.m. every day, the symbol _ means that there is no restriction on feature contents, and *accept* is the action of rule. The detail of time code can be shown in Formulas (38) and (39).

Once the gateway loads this rule, it allows only packets that conform to the rule to pass through, with the rest discarded by default.

Generally speaking, security rules are a kind of language which can be loaded by a security interconnection gateway. The security interconnection gateway has a large capacity and high efficiency of execution to meet the requirements of fine-grained security control of the integrated network of air–ground in multiple domains and achieve the purpose of in-depth protection. In this paper, we propose a multi-category of joint protection security rules.

Security rules have life cycle management, which consists of the generation state of security rule, distribution state, and execution state. The elements mentioned above form the framework of fine-grained multi-domain network interconnection security control, FMNISCF for short. The specific relationships will be shown in Figure 1.

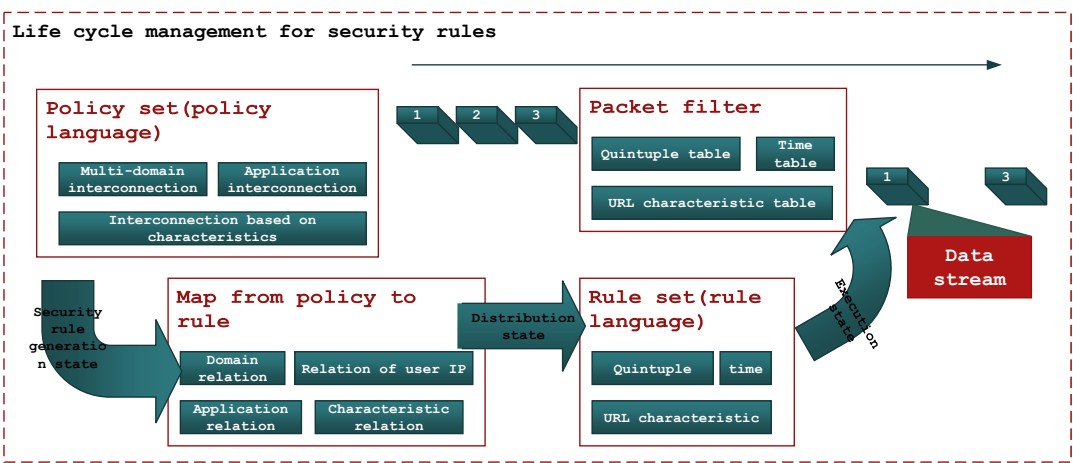

**Figure 1.** FMNISCF: Fine-grained Multi-domain Network Interconnection Security Control Framework.

*2.1. Security Policy*

Security policies in the system are used to help engineers configure access control logic information for multi-domain networks. Therefore, the security policy should be able to represent the information such as domain, application, connect/reject etc, and organize them together in the form of a list to form an overall control map. We design a policy language, which is a collection of strings called $\mathcal{P}$. The semantics of security policies can be expressed as a map:

$$pf : \mathcal{D} \times \mathcal{D} \times APP \rightarrow action, \tag{1}$$

where $\mathcal{D}$ is a set of domains of the network, $APP$ is a set of applications of network and *action* is a set of actions that gateway can do. From the perspective of relationship, in the multi-domain network, application communication among domains needs to rely on IP address, protocol, and port at the network layer. In addition, we also add time and URL (Uniform Resource Locator) characteristics. They form a relationship in a multi-domain network, whose detail is as follows:

$$R_{mdr} \subseteq \mathcal{D} \times \mathcal{U} \times IP \times APP \times PROTOCOL \times PORT \times TIME \times REG, \tag{2}$$

where $\mathcal{U}$ is a set of users in the multi-domain network, $IP$ is a set of IP addresses, $PORT$ is a set of communication port in the network, $TIME$ is a set of time, and $REG$ is a regular expression language.

*2.2. Security Rule*

Security rules can be loaded by the gateway. Corresponding to the policy, we also design a rule language, which is a collection of strings called $\mathcal{R}$. In fine-grained access control logic, it is a map that maps dimensional parameters to actions. It can be shown as follows:

$$rf : \mathcal{T}_1 \times \mathcal{T}_2 \times ... \times \mathcal{T}_n \rightarrow action, \tag{3}$$

where $\mathcal{T}_i$ is a dimension in fine-grained access control logic. In the rule which is designed by engineering personnel, the gateway analyzes packets from seven dimensions: source IP address, destination IP address, protocol type, source port, destination port, gateway time, and URL characteristics.

*2.3. Execution of Packet Filter*

During the process of gateway loading rules, the loader acts as a map. This map transforms a specific rule string into a packet filtering machine. This machine is the semantics that rules represent. It can be shown as follows:

$$ef : \mathcal{R} \rightarrow \mathcal{M}_{rf}, \tag{4}$$

where $\mathcal{M}_{rf}$ is a machine that can load security rules and filter packets. It has two kinds of instructions: *load security rule* and *fiter packet*. The machine can be understood as having a matching tree. It updates the matching tree structure by executing the *load security rule* instruction. When it performs the *fiter packet* instruction, it determines whether the packet is accepted or not by comparing each field of the packet in the matching tree.

*2.4. Map*

The map from policy to rule is a translator that translates the policy language into the rule language. The semantic consistency between the two is required in this process. It can be shown as follows:

$$mf : \mathcal{P} \rightarrow \mathcal{R}. \tag{5}$$

The above is the definition of FMNISCF four components, which are independent of each other and complete the whole life cycle management of security rules through mutual cooperation. In the integrated air-ground multi-domain network, the network communication can be configured according to the actual security requirements of the domain, and the process can be executed safely and efficiently.

**3. Implementation**

In this section, we will use Backus–Naur Form (BNF) paradigm to give a normalized policy/rule language definition and define a map method from policy language to rule language. As a formal description of policies and rules, subsequent security proofs depend on their structure.

*3.1. Security Policy Grammar*

In the following production, the string is enclosed in double quotation "", the brackets [] indicate that the element appears at most once, the asterisk $^*$ indicates that the element appears 0 time or more times, and the parentheses $\langle \rangle$ indicate that the element is a production reference as a whole:

$$policy\ set \quad :: \quad [\text{"}default\text{" " : " } action]\ policy^*, \tag{6}$$

$$
policy \quad :: \quad
\begin{aligned}
& \langle white\ list \rangle \\
& | \quad \langle black\ list \rangle \\
& | \quad \langle scope\ action\ list \rangle,
\end{aligned}
\tag{7}
$$

$$white\ list \quad :: \quad \text{"}white\ list\text{" " : " "\{" } \langle list\ body \rangle \text{ "\}",} \tag{8}$$

$$black\ list \quad :: \quad \text{"}black\ list\text{" " : " "\{" } \langle list\ body \rangle \text{ "\}",} \tag{9}$$

$$scope\ list \quad :: \quad \text{"}scope\text{" "\{" } \langle list\ body \rangle \text{ "\}",} \tag{10}$$

$$,list\ body \quad :: \quad config^*, \tag{11}$$

$$,config \quad :: \quad ID \text{ " } - \text{ " } ID \text{ "," } application \text{ ";"} \tag{12}$$

$$
application \quad :: \quad
\begin{aligned}
& ID \\
& | \quad \text{"[" } \langle app\ list \rangle \text{ "]",} \\
& | \qquad *
\end{aligned}
\tag{13}
$$

$$app\ list \quad :: \quad ID\ (\text{ "," } ID\ )^*, \tag{14}$$

$$action \quad :: \quad \text{"}accept\text{"} \mid \text{"}drop\text{"}, \tag{15}$$

where *ID* is a string literal and its detailed definition can refer to the string definition in the c99 standard.

　　In addition, the functional-independent annotation syntax is not mentioned above. Strictly speaking, they are also part of the policy language.

*3.2. Security Rule Grammar*

$$rule\ set \quad :: \quad [default : action]\ rule^*, \tag{16}$$

$$rule \quad :: \quad \langle ip\ section \rangle\ \text{","}\ \langle dst\ ip\ rule \rangle, \tag{17}$$

$$dst\ ip\ rule \quad :: \quad \begin{array}{l} \langle ip\ section \rangle\ \text{","}\ \langle protocol\ type\ rule \rangle \\ \mid \quad \text{"\{"}\ \langle dst\ ip\ list \rangle\ \text{"\}"} \end{array}, \tag{18}$$

$$protocol\ type\ rule \quad :: \quad \begin{array}{l} section\text{","}\ \langle src\ port\ rule \rangle \\ \mid \quad \text{"\{"}\ \langle protocol\ type\ list \rangle\ \text{"\}"'} \end{array} \tag{19}$$

$$src\ port\ rule \quad :: \quad \begin{array}{l} section\text{","}\ \langle dst\ port\ rule \rangle \\ \mid \quad \text{"\{"}\ \langle src\ port\ list \rangle\ \text{"\}"}\ ' \end{array} \tag{20}$$

$$dst\ port\ rule \quad :: \quad \begin{array}{l} section\text{","}\ \langle time\ rule \rangle \\ \mid \quad \text{"\{"}\ \langle dst\ port\ list \rangle\ \text{"\}"'} \end{array} \tag{21}$$

$$time\ rule \quad :: \quad \begin{array}{l} section\text{","}\ \langle characteristic\ rule \rangle \\ \mid \quad \text{"\{"}\ \langle time\ list \rangle\ \text{"\}"} \end{array}, \tag{22}$$

$$characteristic\ rule \quad :: \quad \begin{array}{l} \langle characteristic\ section \rangle\ \text{" : "}action\text{";"} \\ \mid \quad \text{"\{"}\ \langle characteristic\ list \rangle\ \text{"\}"}\ ' \end{array} \tag{23}$$

$$dst\ ip\ list \quad :: \quad (ip\text{","}\ \langle protocol\ type\ rule \rangle\ )^*, \tag{24}$$

$$protocol\ type\ list \quad :: \quad (section\text{","}\ \langle src\ port\ rule \rangle\ )^*, \tag{25}$$

$$src\ port\ list \quad :: \quad (section\text{","}\ \langle dst\ port\ rule \rangle\ )^*, \tag{26}$$

$$des\ port\ list \quad :: \quad (section\text{","}\ \langle time\ rule \rangle\ )^*, \tag{27}$$

$$time\ list \quad :: \quad (section\text{","}\ \langle characteristic\ rule \rangle\ )^*, \tag{28}$$

$$characteristic\ list \quad :: \quad (\langle characteristic\ section \rangle\ \text{" : "}action\ \text{";")}^*, \tag{29}$$

$$ip \quad :: \quad \begin{array}{l} \langle ip\ number \rangle\ \text{"."}\ \langle ip\ number \rangle\ \text{"."}\ \langle ip\ number \rangle\ \text{"."}\ \langle ip\ number \rangle \\ - \langle ip\ number \rangle\ \text{"."}\ \langle ip\ number \rangle\ \text{"."}\ \langle ip\ number \rangle\ \text{"."}\ \langle ip\ number \rangle' \end{array} \tag{30}$$

$$ip\ section \quad :: \quad \begin{array}{l} \text{"["}\ ip\ (\text{","}\ ip)^*\ \text{"]"} \\ \mid \quad ip \end{array}, \tag{31}$$

$$section \quad :: \quad \begin{array}{l} number\text{"}-\text{"}number \\ \mid \quad \text{"}\_\text{"} \end{array}, \tag{32}$$

$$characteristic\ section \quad :: \quad \begin{array}{l} \langle regula\ rexpression \rangle \\ \mid \quad \text{"}\_\text{"} \end{array}, \tag{33}$$

where *number* is a decimal natural number, *ip number* is a decimal number ranging from 0 to 255, and the detailed definition of *regula rexpression* is available in reference [25].

*3.3. Map Policies to Rules*

As a map, this section transforms the conforming policy language into the corresponding rule language. In combination with the actual situation, IP addresses among users are different, and IP addresses domains are also different. Therefore, we can further conclude that the relation has the following properties:

**Definition 1.** $\forall u \in \pi_{\mathcal{U}}(R_{mdr}). \exists ip. \langle u, ip \rangle \in \pi_{\mathcal{U},IP}(R_{mdr}),$

**Definition 2.** $\forall d \in L(config - ID). \pi_{\mathcal{U}}(\sigma_{\mathcal{D}=d} R_{mdr}) \neq \varnothing,$

**Definition 3.** $\forall app \in L(application - ID). \pi_{APP}(\sigma_{APP=app} R_{mdr}) \neq \varnothing,$

**Definition 4.** $\forall reg_1, reg_1 \in \pi_{REG}(R_{mdr}). reg_1 = reg_2 \vee \mathcal{L}(reg_1) \cap \mathcal{L}(reg_2) = \varnothing,$

**Definition 5.** $\forall \langle d_1, ip_1 \rangle, \langle d_2, ip_2 \rangle \in \pi_{\mathcal{D},IP}(R_{mdr}). d_1 \neq d_2 \longrightarrow ip_1 \neq ip_2,$

**Definition 6.** $\forall \langle u_1, ip_1 \rangle, \langle u_2, ip_2 \rangle \in \pi_{\mathcal{U},IP}(R_{mdr}). u_1 \neq u_2 \longrightarrow ip_1 \neq ip_2,$

where $\pi$ is a projection operation on a relationship. The parameters between them are the terms to be projected. The function $L()$ represents a collection of strings that a node of grammar tree matches. The function $\mathcal{L}()$ is a set of strings represented by a regular expression. Now, we describe the steps of the algorithm by means of the following examples:

$$
\begin{bmatrix}
default & : & accept \\
white\ list & : & \{A-B, app1\ A-C, [app1, app2]\ A-B, app2\} \\
black\ list & : & \{A-B, app1\ A-B, app3\} \\
scope\ list & : & \{A-B, app1, app2, app3\}
\end{bmatrix}. \tag{34}
$$

1. From Formulas (6) and (16), we can know that $\varepsilon \in policy\ language \wedge \varepsilon \in rule\ language$, where $\varepsilon$ is empty string. We specify that, when the policy is $\varepsilon$, the corresponding rule image is $\varepsilon$. Using the same policy, we transform $["default"\ ":"\ action]$ statement from the policy to rule.
2. Transform a *policy* statement into a *rule* statement. We divide the *policy* into three cases: when the *policy* is *whitelist*, the rule action is translated as "*accept*"; when the policy is the *blacklist*, it is translated as "*drop*", and, in the last case, it is translated as the default statement of the opposite action.

   First, merge every *config* bar in the whitelist into a tree (white list tree). Traversal from root to leaf: if the two nodes currently being compared are the same, merge them into one node; otherwise, the remaining subtrees are merged under the parent node. Figure 2 is the algorithm of *config* in *white list* to merge into white list tree.

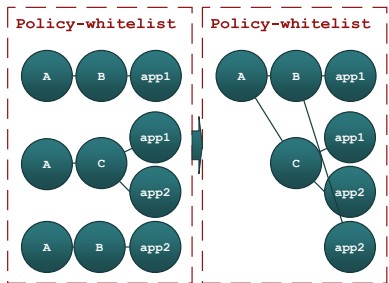

**Figure 2.** Tree mergers of config in the whitelist.

Next, merge each *config* of the blacklist into the white list tree in turn and conflict checking is performed for each merge. The specific process is that, if the exact same subtree appears, the system will prompt for a "detected conflict" message. Figures 3 and 4 show an example of a conflict detected.

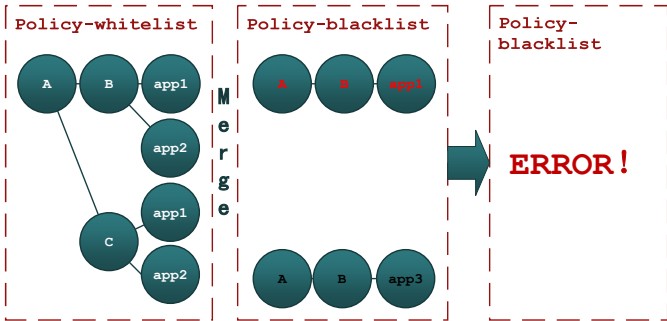

**Figure 3.** Tree mergers of config in the blacklist.

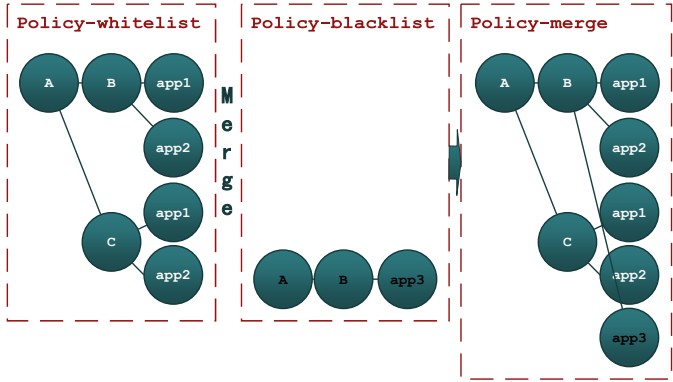

**Figure 4.** Tree mergers of config in the blacklist conflict checking.

Next, each scope list is merged into the whitelist tree, as opposed to the default operation, which merges the list (if the default action is accepted, it merges with the blacklist; otherwise, it merges with the whitelist). Figure 5 shows that the scope list is merged into the whitelist tree.

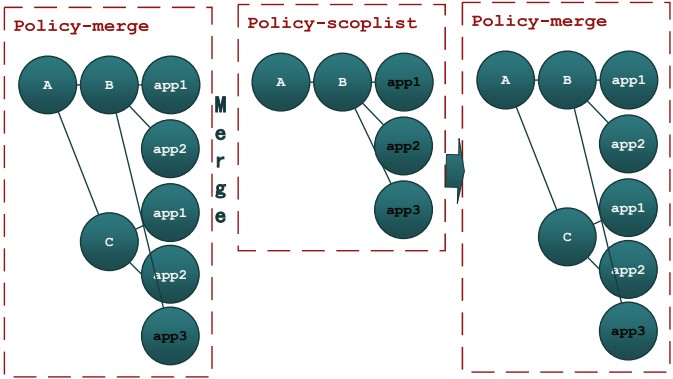

**Figure 5.** Tree mergers of config in the scope list.

Through this algorithm, a new tree can be merged into the existing tree, and the original principle can be maintained. The operation to merge the subtree into the original tree is represented by plus sign $+$.

3.  If the *listbody* is $\varepsilon$, translate $\varepsilon$ as follows (all other parts of the signed $^*$ are the same). When there is only one configuration, from Formula (12), we have:

$$config \quad :: \quad ID\ ``-"\ ID\ ``,"\ application\ ``;".$$

From script (34), we get the domain set $\{A, B, C\}$ and the application set $\{app_1, app_2, app_3\}$. For example, the domain of $A$ corresponds to IP interval [10.0.0.1-10.0.0.3 10.0.0.5, 10.0.0.7]. Similarly, the IP interval of the domain $B$ is [6,8-9]. For the language element $A - B$ of the security policy script, it can be translated into the source IP interval, and the destination IP interval of the language element in the security rules script respectively. It can be written as [1-3,5,7,10],[6,8-9]. Similarly, $a - c$ can be translated into [1-3,5,7,10],[10,12].

4.  Ignoring the control characters "," and ";", we consider the second half of the formula "*application*". We mark these *APP*s as $app_i$. By Definition 3, for each *ID*, it can obtain the protocol type, destination port, time, and characteristics via relational operation expression $\pi_{PROTOCOL,PORT,TIME,REG}(\sigma_{APP=app_i}(R_{mdr}))$. The source port is used for its own side to receive messages and is randomly assigned with no restrictions. Covert this relational structure into a tree structure in order. The application element is translated into protocol type interval, source port interval, destination port interval and time interval characteristics in security rules. By searching for application name in scripts, we can get a security rule subtree with an application from the multi-domain network. For example, $app_1$ uses TCP, UDP and ICMP protocols. Among them, TCP uses port 80 as the destination port, and UDP uses port 90 as the destination port. The usage time of app1 is limited to 5:00 a.m.–11:00 a.m. per day. The algorithm is illustrated by the following Figure 6:

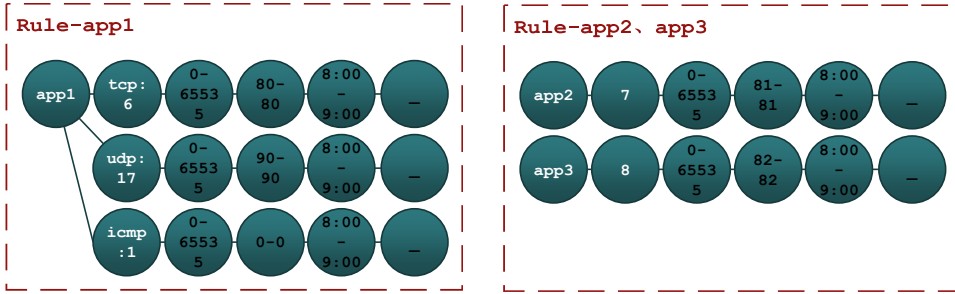

**Figure 6.** Transform from relationship to application match tree.

5.  In steps 3 and 4, multiple tree merges are involved. Our approach is the same merge, different splits. Combining the whitelist tree of step 2, the security policy language is finally translated into the security rules script. In this process, the subtrees of the step 2 are connected, and the corresponding actions are added to the final rule part in combination with the black and white list action of the whitelist tree.

In the implementation, the packet filter module will call the mutual exclusion of the two operations through the synchronization variable *lock* when loading the forest rules or filtering the packet. Algorithm 1 is described precisely as follows:

---

**Algorithm 1:** Fine-grained packet filter

---

**Input:** Packet *t*, Forest rules *rt*

**Output:** Action *r*

1  initialization: Synchronous *lock*, int *status*;
2  P(*lock*);
3  **if** *status==LOAD_RULES* **then**
4      parse_policies(*rt*);
5      merge trees into forest (*Algorithm 2*);
6      map policies to rules (*Algorithm 3*);
7      convert absolute Time to relative time (*Algorithm 4*);
8  **else**
9      **if** *status==FILTER_PACKET* **then**
10         update time interval (*Algorithm 5*);
11         V(*lock*);
12         return *r* = filtering_packet(*t*);
13     **end**
14 **end**
15 V(*lock*);
16 return NOACTION;

---

We define a type named *Forest_policy* whose type can be seen in Formula (35):

$$
\begin{aligned}
&Forest\_policy\{ \\
&\quad Forest\_policy\ parent; \\
&\quad List < Forest\_policy > child\_list; \\
&\quad Union\{List < Interval > data\_list; \\
&\quad\ Enumerate\ action; \\
&\quad \}list\_action; \\
&\},
\end{aligned}
\tag{35}
$$

where *parent* is the parent of this level, and *child_list* is its children. *child_action* is the data of the node.

There are four levels of the forest which has been built. They are the source of the domain, the destination of the domain, the application, and the action of the forest policy, respectively. Corresponding to *Forest_policy*, we also define *Forest_rule*. There are eight levels that are the source of IP, the destination of IP, the source of the port, the destination of the port, the protocol type, the time, the characteristic, and the action. See Formula (36):

$$
\begin{aligned}
&Forest\_rule\{ \\
&\quad Forest\_rule\ parent; \\
&\quad List < Forest\_rule > child\_list; \\
&\quad Union\{List < Interval > data\_list; \\
&\quad\ Enumerate\ action; \\
&\quad \}list\_action; \\
&\}.
\end{aligned}
\tag{36}
$$

The policies can be mapped to the rules by Algorithm 3. The initialization of it also needs to be finished by Algorithm 2. It merges each policy tree in the policy file into the forest policy. In the process, the algorithm simultaneously merges the policies and detects conflicts.

---

**Algorithm 2:** Merge trees into forest

**Input:** Single policy $p$, forest policies $f$
**Output:** forest policies $f$

1 **for** $i = 0$; $i < f.child\_list.size()$; $i$++ **do**
2    $f\_root_i = f.child\_list.get(i)$;
3    **if** $f\_root_i == t$ **then**
4      **for** $j = 0$; $j < f\_root_i.child\_list.size()$; $j$++ **do**
5       $f\_node_j = f\_root_i.child\_list.get(j)$;
6       $p\_node = p.child\_list.get(0)$;
7       **if** $f\_node_j == p\_node$ **then**
8        **for** $k = 0$; $k < p\_node.child\_list.size()$; $k$++ **do**
9         $p\_app_k = p\_node.child\_list.get(k)$;
10         **for** $l = 0$; $l < f\_node_j.child\_list.size()$; $l$++ **do**
11          $f\_app_l = f\_node_j.child\_list.get(l)$;
12          **if** $f\_app_l == p\_app_k$ **then**
13           **if** $f\_app_l.action == SCOPE$ **then**
14            **if** $p\_app_k.action == BLACK$ *or* $WHITE$ **then**
15             $f\_app_k.action = p\_app_l.action$;
16            **end**
17           **end**
18          **else**
19           **if** $f\_app_l.action! = SCOPE$ **then**
20            **if** $p\_app_k.action! = SCOPE$ **then**
21             throw "conflict detection!";
22            **end**
23           **end**
24           **if** $f\_app_l.action! = SCOPE$ **then**
25            $f\_app_k.action = p\_app_l.action$;
26           **end**
27          **end**
28         **end**
29         $f\_app_k.child\_list.add(p\_app_l)$;
30        **end**
31       **end**
32      **end**
33      **return** $f\_root_i.child\_list.add(p\_node)$;
34     **end**
35 **end**
36 **return** $f.child\_list.add(p)$;

---

Each policy $p$ in the list is regarded as a tree, so policies $f$ can be considered as a forest. Both forest and trees are three-layer structures. The layers represent source domain, destination domain, and app, respectively. As for app nodes, they all contain an action field ($BLACK/WHITE/SCOPE$). The function of Algorithm 2 is to merge trees into a forest.

Algorithm 2 is described precisely as follows:

Travel through every tree in the forest and try to merge policy $p$ into the tree. If it fails to merge policy $p$ into any tree in the forest, just add policy $p$ into $f.child_list$. The method of merging trees is as follows. First, compare the root node of policy $p$ with that of the $i$-th tree in the forest, which are denoted as $t$ and $f\_root_i$, respectively. If they are different, policy $p$ cannot be merged into the $i$-th tree.

Try to merge it into another tree. If they are the same, compare their child nodes which are denoted as *p_node* and *f_node*.

If a child node of $f\_root_i$ denoted as $f\_node_j$ is the same as *p_node*, compare their child nodes which are denoted as *f_app* and *p_app*, or add *p_node* to $f\_root_i.child\_list$.

For each *p_app*, if there is no *f_app* like it, add it to $f\_node_j.child\_list$. If there is an *f_app* denoted as $\_app_l$ which is the same as $p\_app_k$, compare their actions. If one is *SCOPE* and the other is *BLACK* or *WHITE*, then $f\_app_k.action = BLACK$ or *WHITE*. If they are different and neither is *SCOPE*, throw "conflict detection!".

Algorithm 3 is described precisely as follows:

In Algorithm 3, it maps forest policies $f$, $p$ into forest rules $r$, $t$. For each tree in the forest $f$, $p$, use the following method to turn it into a rule tree. As mentioned above, the trees of policy forest are three-layer structure, representing source domain, destination domain and app, respectively. Thus, given the root of the *i*-th tree, we can get the source domain, and further obtain the source IP address $srcip_i$ by searching the set of relationships $R_{mdr}$. The destination IP address $dstip_j$ can be obtained similarly. Then, add it to $srcip_i.child\_list$. From the app, we can get the information of protocol, source port, destination port, time, reg and action. Add them to the *child_list* of their previous element respectively and make *reg* the leaf node of the rule tree. Leaf nodes of rule trees contain an action field which is the same as the app node of the original policy tree.

---

**Algorithm 3:** Map policies to rules

**Input:** Forest policies $fp$
**Output:** Forest rules $rt$

1 initialization: $fp$.init(), $rt$.init();
2 **for** $i = 0; i < fp.child\_list.size(); i++$ **do**
3    $root = fp.child\_list$.get($i$);
4    $srcip_i = \pi_{IP}(\sigma_{D=root}R_{mdr})$;
5    **for** $j = 0; j < root.child\_list.size(); j++$ **do**
6       $node = root.child\_list$.get($j$);
7       $dstip_j = \pi_{IP}(\sigma_{D=node}R_{mdr})$;
8       $srcip_i.child\_list$.add($dstip_j$);
9       **for** $k = 0; k < node.child\_list.size(); k++$ **do**
10          $app = node.child\_list$.get($k$);
11          $protocol_k = \pi_{PROTOCOL}(\sigma_{APP=app}R_{mdr})$; $dstip_j.child\_list$.add($protocol_k$);
12          $srcport_k = [0\text{-}65535]$;
13          $protocol_k.child\_list$.add($srcport_k$);
14          $dstport_k = \pi_{PORT}(\sigma_{APP=app}R_{mdr})$;
15          $srcport_k.child\_list$.add($dstport_k$);
16          $time_k = \pi_{TIME}(\sigma_{APP=app}R_{mdr})$;
17          $dstport_k.child\_list$.add($time_k$);
18          $reg_k = \pi_{REG}(\sigma_{APP=app}R_{mdr})$;
19          $time_k.child\_list$.add($reg_k$);
20          $leaf_k = app.action$;
21          $reg_k.action = leaf_k$;
22       **end**
23    **end**
24    $rt.list$.add($srcip_i$);
25 **end**

---

Time coding is divided into absolute time and relative time. Time elements such as year, month, day, week, hour, minute and second are used to encode relative time. Absolute time is encoded in seconds that have elapsed since 1 January 1970. The relative time of the base structure from low to high is 60, 60, 24, 7, 31, and 12. Of these, number 7 means that the week is unlimited. Number 31 means days are unlimited. Number 12 means month is unlimited. For the time format "$yyyy - mm - ddhh : MM : ss$",

$$\begin{aligned} \text{intToRe}(mm, dd, w, hh, MM, ss) = ss \times 1 + MM \times 60 + hh \times 60 \times 60 + w \times 24 \times 60 \times 60 \\ + dd \times 7 \times 24 \times 60 \times 60 + mm \times 31 \times 7 \times 24 \times 60 \times 60, \end{aligned} \tag{37}$$

where the week $w$ can be calculated using the library function provided by "time.h". For example, from 9:00 a.m. to 10:00 a.m. every day, the code of relative time is Formulas (38) and (39):

$$287485200 = \text{intToRe}(12, 31, 7, 9, 0, 0), \tag{38}$$

$$287488800 = \text{intToRe}(12, 31, 7, 10, 0, 0). \tag{39}$$

If it's 0:00 a.m. to 0:00 a.m. per day, this means that there's no limit about time. For "$\text{intToAb}(\cdot)$", we first reverse calculate $[mm, dd, w, hh, mm, ss] = \text{intToRe}^{-1}(\cdot)$, obtain the current absolute time via "$time(\cdot)$", then assign the rest to the absolute time structure, and finally calculate the absolute time value through the library function "$mktime(\cdot)$".

Algorithm 4 is described precisely as follows.

In Algorithm 4, given the relative time interval which begins with $stamp_1$ and ends with $stamp_2$, the corresponding absolute time interval can be obtained. Time stamp is an integer and can be converted into relative time or absolute time through function "$\text{intToRe}(\cdot)$" or "$\text{intToAb}(\cdot)$". Using function "$\text{intToRe}^{-1}(\cdot)$" for $stamp_1$ and $stamp_2$, we can get six pairs of numbers representing the restrictions on month, date, day of week, hour, minute, and second. Placing restrictions on month means the period from one month to one month every year and no restriction implies January to December. Similarly, for date, day of week, hour, minute, and second, whether to make restrictions or not is optional.

Using function intToAb() for update, the time stamp of current time, we can get the current month, date, day of week, hour, minute, and second. If current time meets the limits to month, date, and day of week, there are two cases to consider. One case is that there are no limits to hour, minute, and second, and the other is that hour, minute, and second are limited. For the former, the absolute time interval converted is from 0:0:0 a.m. that day to 0:0:0 a.m. the next day. For the latter, it is the limited time interval that day. If current time does not meet any limit to month, date or day of week, the absolute time interval converted is from 1 January 1970, 0:0:0 a.m. to 1 January 1970, 0:0:0 a.m..

Algorithm 5 is described precisely as follows:

Since absolute time interval are converted from relative time interval according to current time, the absolute time interval converted should change as current time changes. Thus, it is necessary to invoke Algorithm 5 to update absolute time interval to ensure accuracy. The update input is the time stamp of the last time calling Algorithm 4. If the time stamp of packet *packetTime* is bigger than *update*, invoke Algorithm 4 and update *update* ensuring that the next update will be completed before the time constrained.

---

**Algorithm 4:** Convert relative time into absolute Time

**Input:** int $stamp_1$, $stamp_2$
**Output:** int $update$, $ustamp_1$, $ustamp_2$

1   $update$ = time();
2   $[mon_1,mday_1,wday_1,hour_1,min_1,sec_1]$ = intToRe($stamp_1$);
3   $[mon_2,mday_2,wday_2,hour_2,min_2,sec_2]$ = intToRe($stamp_2$);
4   $[mon_a,mday_a,wday_a,hour_a,min_a,sec_a]$ = intToAb($update$);
5   **if** $mon_1 <= mon_a <= mon_2$ || $(mon_1==mon_2==13)$ **then**
6     **if** $mday_1 <= mday_a <= mday_2$ || $(mday_1==mday_2==8)$ **then**
7       **if** $wday_1 <= wday_a <= wday_2$ || $(wday_1==wday_2==32)$ **then**
8         **if** $hour_1==min_1==sec_1==hour_2==min_2==sec_2==0$ **then**
9           $ustamp_1 = ustamp_2$ = intToAb(0,0,0);
10         **else**
11           $ustamp_1$ = intToAb($hour_1,min_1,sec_1$);
12           $ustamp_2$ = intToAb($hour_2,min_2,sec_2$);
13         **end**
14         return $[update, ustamp_1, ustamp_2]$;
15       **end**
16     **end**
17 **end**
18 $ustamp_1 = ustamp_2$ = intToAb(1970,1,1,0,0,0);
19 return $[update, ustamp_1, ustamp_2]$;

---

**Algorithm 5:** Update time interval

**Input:** int $packetTime$, $update$, $stamp_1$, $stamp_2$
**Output:** int $update$, $ustamp_1$, $ustamp_2$

1   **if** $packetTime >= update$ **then**
2     $[ustamp_1, ustamp_2]$ = Algorithm 4($stamp_1$, $stamp_2$);
3     h = stamp(0,0,0) + 24*60*60 + rand()%(h*60*60);
4 **end**

---

*3.4. Match Tree Building of Packet Filter*

The transmitted security rules are built into a packet filter match tree in the gateway machine. The implementation method is to build an 8-layer filter matching tree. The root node is in the first layer, the second layer is the source IP range of child nodes, and the third layer is the source IP range of which the parent node is IP range node. By analogy, the fourth floor is protocol type interval, the fifth layer is the source port interval, the sixth layer is the destination port, the seventh layer is the time interval, and the eighth floor is URL characteristics. As you can see from step 2, the security rule script has a very similar structure to the filter match tree. This procedure can be understood as the gateway machine $\mathcal{M}_{rf}$ executing the *load security rule* instruction.

After the matching tree has been built, when the gateway machine executes the instructions *filter packet*, it will start from the root node, from top to bottom, and select the matching subtree from the eligible subtree, until it finds the action of the leaf node. In particular, if no eligible subtrees are found at a certain level, it jumps out of the tree traversal and performs the default action.

## 4. Security of System

We treat the *policy set* as a string of elements, which can be abstracted into three categories: Empty set $\varepsilon$, $default$" : "$action$, and $policy$. According to the policy set grammar, we know that:

**Definition 7.** *1.    Empty set is a policy set.*
*2.    The default action is a policy set.*
*3.    A policy set join policy is a policy set.*
*4.    The other cases are not a policy set.*

The join operation between the policy set and the policy is marked as $\oplus$. According to the above definition, the join operation is closed on *policy set*. The $\langle policy\ set\ ,\oplus \rangle$ now constitutes an algebraic system.

In the same way, rule sets are also divided into three similar parts: empty set $\varepsilon$, $default" : "action$, and *rule*.

According to the rule set grammar, we know that:

**Definition 8.** *1.    Empty set is a rule set.*
*2.    The default action is a rule set.*
*3.    A rule set join rule is a rule set.*
*4.    The other cases are not a rule set.*

The join operation between the rule set and the policy is marked as $\otimes$. According to the above definition, the join operation is closed on *rule set*. The $\langle rule\ set\ ,\otimes \rangle$ now constitutes an algebraic system.

*4.1. Map Security*

Review Section 3.3 map policies to rules, we stipulate map $mf$:

$$
mf(p) = \begin{cases} \varepsilon & p = \varepsilon, \\ default" : "action & p = default" : "action, \\ mf(p') + p_{config} & p = p'p_{config}. \end{cases} \tag{40}
$$

The meaning of the operator $+$ is specified above.

**Theorem 1.** $\forall p \in policy\ set.\ \exists r \in rule\ set.\ mf(p) = r$

**Proof.** From Formula (40), if $p = \varepsilon$ or $p = default" : "action$, the theorem is established.

Assuming that $mf(p')$ satisfies the Theorem 1, we can deduce that this policy has a corresponding $tree_{p'}$. From Step 3, the policy $p_{config}$ also has a corresponding $tree_{p_{config}}$. From Step 5, these trees can be merged into a $tree_p$, which can be translated into a rule.

Now, this theorem is proved by induction.    □

From Theorem 1, we can reach a conclusion that, for every set of grammatical and semantical policies in the network, we can always find the set of rules corresponding to them.

*4.2. Semantic Security*

In terms of the semantics of rules, machine $M_{rf}$ with the same function is a kind of rule class for different forms of rules. The difference of these rules can be reflected in the choice of the interval, such as the case of $1-3, 3-4$ and $1-2, 2-4$. However, in terms of the point of view of the packet filtering function of the machine, they are the same.

**Lemma 1.** $!\exists p \in policyset, M_{pf1} \neq M_{pf2}.\ ef(mf(p)) = M_{pf1} \wedge ef(mf(p)) = M_{pf2}.$

**Proof.** Suppose there is a rule $p$ that performs both receive and discard actions for a packet. This requires the encoding of the interval to have two meanings, which does not conform to the

natural number encoding, time encoding, and collection of characteristic encoding (from Definition 4). This leads to a contradiction.  □

**Theorem 2.** *The map ef is well-defined. (It involves semantics of mf)*

**Proof.** According to Lemma 1, we know that the relation of the semantics of policy/rule forms a partition of it. From Theorem 1, we can know $\forall p \in policy\ set. \exists M_{pf} = ef(mf(p))$. Combined with the properties of partition, if two $M_{pf1}$, $M_{pf2}$ are different, the corresponding partition must be different. This satisfies two requirements of a well-definition.  □

By Theorem 2, we can guarantee that the "Execution of packate filter" component will not generate ambiguity, thus ensuring semantic security.

*4.3. Optimization*

**Theorem 3.** *Nodes in the same hierarchy can be arranged in order except for their characteristics.*

**Proof.** Considering the source IP, destination IP, protocol type, source port, destination port and time, they are all encoded with a natural number, and there is no intersection between the intervals of each node, so they can be arranged orderly.  □

By Theorem 3, we can optimize the packet filtering algorithm. In addition to characteristic nodes, nodes at other levels can be arranged in an orderly manner, so the gateway can use the binary search method to retrieve the rule tree.

## 5. Performance

The operation interface of the three procedure in the framework FMNISCF is shown as follows:

Figure 7a shows the configuration of a security policy on a web browser. The newly added security policy is saved in the database, and then the user can see it in the list. Figure 7b shows the deployment of security rules to the gateway through a web browser. While clicking the "Distribute" button, the system automatically converts security policies into security rules and deploys them to the gateway. Figure 7c shows the gateway loading security rules sent by the visual management system into the gateway through the security rules loader. Thus, the interconnected security gateway can filter fine-grained packets.

We test the performance of three processes of the framework FMNISCF. Since instructions are distributed by socket communication, the selection of operating system version does not affect FMNISCF deployment. The test hardware environment, software environment and running speed under different configurations are given below:

Table 1 shows the interconnected security gateway and the hardware environment of the visual management system. Interconnection security gateway, controlled by the visual management system, can accept or drop the packets passing through the network according to the instructions sent by the system.

We use software "iPerf3" [26] as the packet sending tool. It simulates communication scenarios in the network.

Table 2 shows the software environment of each state during the rule life cycle management. In the policy transfer phase, a database is required to store security policies, security rule data, and other management information. The data caching software in the web background improves the speed of data query in the system. Web service programs rely on JVM and rapid development framework spring. Fine-grained web management requires support from Shiro components. In the front-end, JavaScript provides rich support of the running script. In the phase of rule loading and packet filtering, the network modules in the kernel are used to provide support.

After setting up the running environment, we can test the three processes in the framework FMNISCF. The first step is to write the policy configuration file. Transform policies into rules through the map. The following is the rate at which rule profiles are generated under different semantic environments in a multi-domain network.

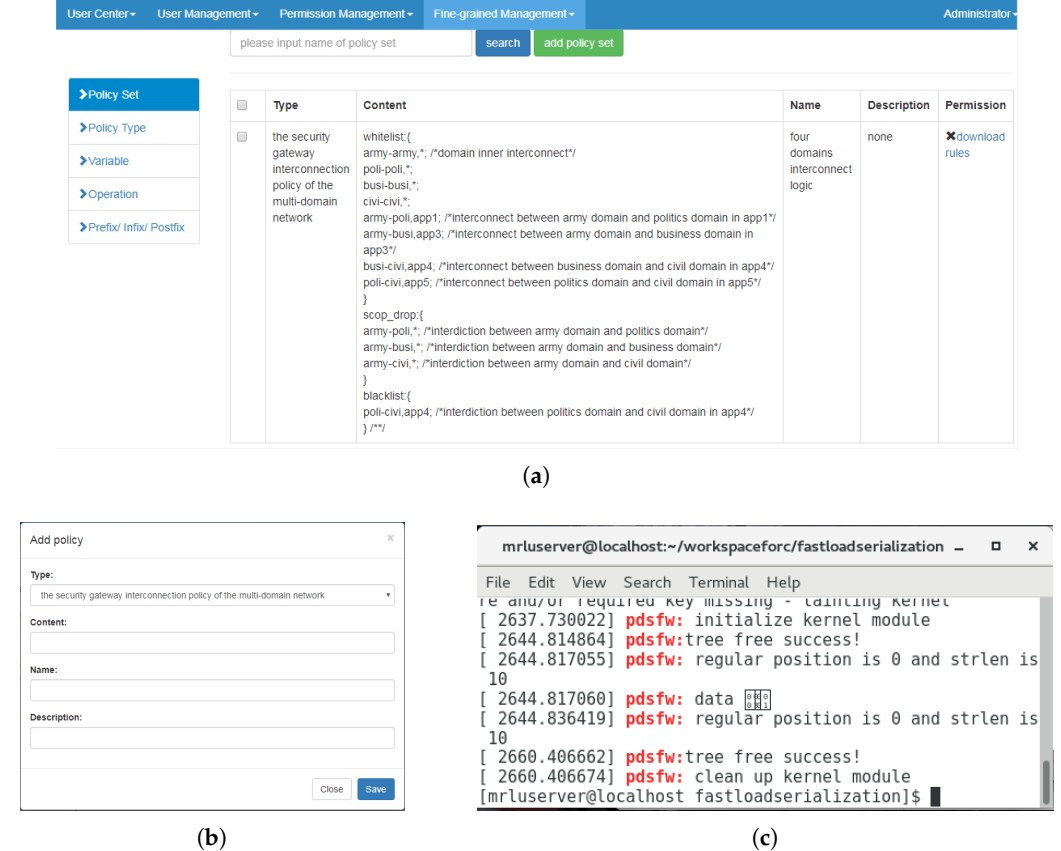

**Figure 7.** Operation interface. (**a**) map from policy to rule and distribution; (**b**) add policy configuration; (**c**) packet filter execution.

**Table 1.** The Relation between test unit and running environment of hardware.

| CPU | DDR4 | OS | Test Unit |
|---|---|---|---|
| Intel(R) Xeon(R) E5-2670 0 2.60 GHz | 128 G | Centos 7 | Policy translation |
| Intel(R) Xeon(R) E3-1220 V2 3.10 GHz | 4 G | Ubuntu 14 | Rule loading Packet filter |

**Table 2.** The relation between test unit and running environment of FMNISCF.

| Software | Version | Test Unit |
|---|---|---|
| database | MySQL 5.0 | Policy translation |
| in-memory data structure store | redis 4.0 | Policy translation |
| JVM | JDK1.8.0 | Policy translation |
| web framework | spring MVC 4.2 | Policy translation |
| persistence framework | mybatis 4.2 | Policy translation |
| Java security framework | Apache Shiro 1.2.5 | Policy translation |
| JavaScript library | JQuery 1.8.3 | Policy translation |
| netfilter | linux kernel 3.10.0-862.el7.x86_64 | Rule loading Packet filter |
| iPerf3 | iperf-3.1.3 | Packet sending |

In Figure 8, the time of generating security rules increases linearly with the number of policies. When the number of policies reaches 660,000, generating security rules only costs about 5 s. This means that, in actual deployments, the system can support very complex security policy configurations and keep the rule generation process efficient.

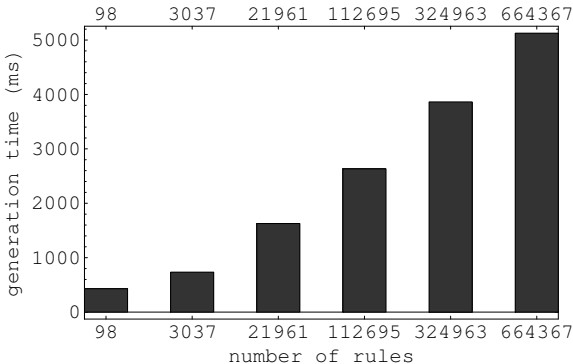

**Figure 8.** Time of rule generation.

Next is the test of time of the rule configuration file loading. In Figure 9, when the number of security rules is 400,000, the loading time of the interconnected security gateway is about 9 s, which means that, in the actual deployment, the system can quickly load the rules generated by the system and respond to the changes in the network. In different rule scales, the loading time changes are shown as follows:

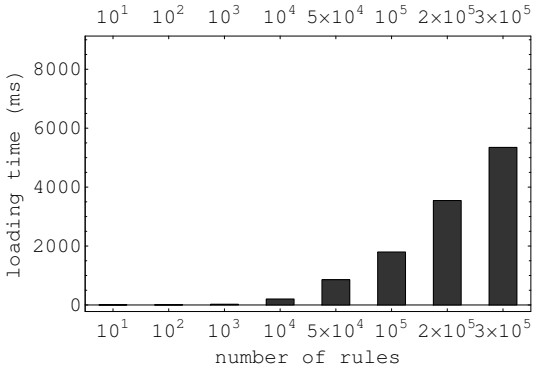

**Figure 9.** Time of rule loading.

Finally, the iPerf3/pktgen generates packet traffic to test the throughput of gateway after loading rules. In the experiment, the tools are used to generate 10 Gbps of traffic. We compare the iptables, DPDK-IPFW, BSD-IPFW, BSD-pfsense, and FMNISCF.

In Figure 10, we analyze the data: as for the optimization of network card drive, DPDK-IPFW can increase the limit value of the data transmission speed of the device. In the actual transmission, it reaches 9.3 Gbps. Furthermore, DPDK-IPFW supports up to 23,207 rules. In addition, BSD-IPFW supports up to 65,535 rules and maintains a stable transmission speed of 1.03 Gbps when the number of rules is greater than 1 W. This shows that BSD-IPFW is very suitable for deployment on the network equipment with a Gigabit network card. This speed is just the upper limit which is generally supported by the intermediate nodes of the network. The network speed of BSD-pfsense can only maintain at 0.2 Gbps. We compare other schemes with ours. It can be seen from the comparison that the throughput of other schemes decreases rapidly with the increase of rules. The speed of other schemes has dropped to 1 Gbps when there are 1 W rules. In contrast, when the number of security rules is 50 W, the throughput of FMNISCF can still maintain 4.6 Gbps. This means that, in the actual deployment, even if the system is loaded with complex filtering rules, the network data can still be transferred quickly. Our plan does not lead to a significant reduction.

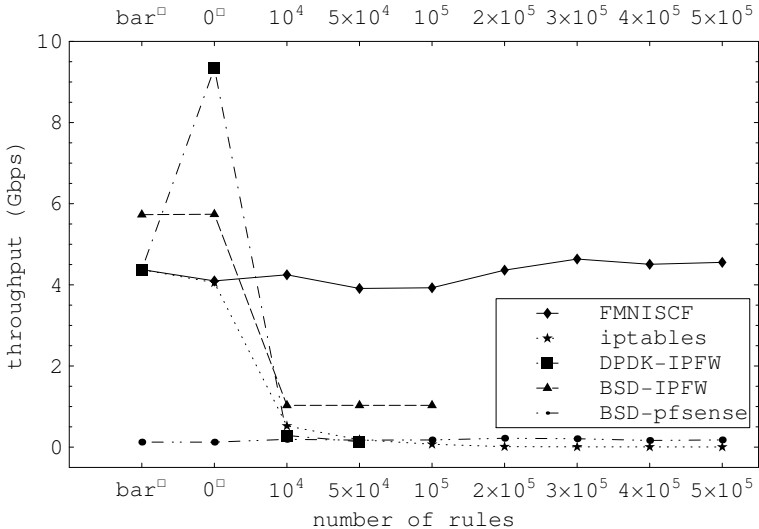

**Figure 10.** Packet throughput of Iptables and ours.

## 6. Conclusions

This paper proposes the framework of a fine-grained multi-domain network interconnection security control, FMNISCF for short. BNF is used to design the grammar specifications of rules and policies. The map method from policy to rule is designed based on the semantics of integrated air-ground multi-domain network. Unifying policies and rules into the structure of the matching tree is the key to establishing the relationship between policies and rules. This structure also provides a scheme for the implementation of packet filtering semantics of gateway. In terms of security analysis, we discuss the security of design about map and semantic. This ensures that the system does not "crash" or generate ambiguities. Combining the characteristics of the system, we optimize the matching tree structure. Finally, we test the system running environment and the running time of each stage, respectively. Our solution is more efficient than iptables in terms of packet filtering performance. The framework can be used to filter more protocols/attacks from the application layer. In some common SQL injection attacks, mining Trojan attacks and other scenarios, it can filter this type of packet in a fine-grained way. In periodic application scenarios, for example, when employees arrive at work at 9:00 a.m. and leave at 5:00 p.m. on weekdays, window time can be set to prevent employees from accessing the company's confidential database after work. Through the establishment of multiple domains, the framework can implement the access control management of each department of the company.

## 7. Future Research Work

In the future, we will design a fine-grained intrusion detection system. Combined with this system, many security policies can be automatically generated by the intrusion detection system and quickly deployed into the FMNISCF. In the intrusion detection system, the attack behavior is analyzed by combining multi-domain network semantics. This includes information such as network topology, department organization structure, and application protocol structure. Coordinated detection and response to attacks are implemented on distributed gateways supported by time synchronization.

**Author Contributions:** Conceptualization, B.L. and R.C.; methodology, B.L. and Y.L.; software, B.L. and L.T.; validation, B.L., L.T. and H.W.; formal analysis, B.L.; investigation, B.L. and R.C.; resources, Y.L.; data curation, B.L., L.T. and H.W.; writing—original draft preparation, B.L., L.T. and H.W.; writing—review and editing, B.L.; visualization, B.L.; supervision, Y.L.; project administration, Y.L.; funding acquisition, Y.L. All authors have read and agreed to the published version of the manuscript.

**Funding:** This research was funded by the National Key R&D Program of China (No. 2016YFB0800302), and funded by the BUPT Excellent Ph.D. Students Foundation (No. CX2019230).

**Acknowledgments:** This research is supported by the National Key R&D Program of China (No. 2016YFB0800302) and the BUPT Excellent Ph.D. Students Foundation (No. CX2019230).

**Conflicts of Interest:** The authors declare no conflict of interest.

## Abbreviations

The following abbreviations are used in this manuscript:

| | |
|---|---|
| FMNISCF | Fine-grained Multi-domain Network Interconnection Security Control Framework |
| TCP | Transmission Control Protocol |
| ODENOS | Object-Defined Network Allocation System |
| SDN | Software Defend Network |
| 2FA | Two-Factor Authentication |
| XACML | eXtensible Access Control Markup Language |
| URL | Uniform Resource Locator |
| IP | Internet Protocol |
| BNF | Backus–Naur Form |
| UDP | User Datagram Protocol |
| ICMP | Internet Control Message Protocol |
| JVM | Java Virtual Machine |
| JDK | Java Development Kit |
| MVC | Model View Controller |
| Gbps | Gigabits/Second |
| DPDK | Data Plane Development Kit |
| BSD | Berkeley Software Distribution |
| IPFW | Internet Protocol Firewall |
| IOT | Internet of Things |

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
