# Peer review of "FMNISCF: Fine-Grained Multi-Domain Network Interconnection Security Control Framework"

_applsci, doi:10.3390/app10010409_

Round 1

Reviewer 1 Report

The paper is interesting, but I see some aspects that might be discussed further. 

Please check the attached PDF where authors can find notes from my review.

I am interested in knowing about the implementation of the time functions and how time is retrieved. Can time be attacked to open wider or close time windows where packets can be accepted or rejected? 

During your tests you create the policies in a CentOS and the rules in a different machine. How this can affect security? Different architectures (if this was the case) can affect how time is encoded? Is the point between these two machines able to suffer from attacks to rewrite rules? 

Why not using some graphical interface as NODERED to create the policies? 

Algorithm 4 and the implementation issues. How time is managed is important. I miss some further discussion on this. 

Check (page 11 the algorithm placements -see notes-)

Screenshots in english, please. Also, they are way too small to read. 

Check figure 8 and 10.

Reviewer 2 Report

This work follows the fine-grainedmulti-domain interconnection security control, generates rules and builds a tree structure of rules to filter packets. In the experimental part, they compared the performance of our scheme with the netfilter/iptables of Linux operating system.

Basically, the work follows the previous studies, but it is unclear how significant the contributions are. The paper structure needs improvement, i.e., the introduction should be better motivated. The evaluation part is not convincing, as it only compares with iptable. More recent studies should be considered, like [9-12] and below.

A fine-grained multi-source measurement platform correlating routing transitions with packet losses. Computer Communications 129: 166-183 (2018)

Reviewer 3 Report

Fine-grained multi-domain interconnection security control is very important within high-traffic network platforms and devices and so various methods have been used to solve the problems related to multi-domain interconnection, conflict detection, rule transformation and fine-grained control. The authors present some of these methods and propose a new framework of a fine-grained multi-domain network interconnection FMNISCF. The framework combined with multi-domain network is based on transformation from security policies to security rules and replaces the traditional manual way of configuring rules. The framework supports the whole life cycle management of rules from generation state, distribution state to execution state. Security and semantic security aspects are analyzed and optimized respectively. A series of experiments are done to verify the efficiency of the scheme.

Conclusions should be extended.

Future research work is missing.
